# PHASE TRANSITIONS IN CONTRASTIVE LEARNING

## ABSTRACT

How do self-supervised models actually train? We study the training dynamics of contrastive learning in three settings: a theoretical linear setting, on a low-dimensional physics-inspired dataset, and on full-fledged computer vision datasets including ImageNet. In all three settings, we show the existence of phases, i.e. locally stable or metastable representations, and of phase transitions, wherein a model rapidly and unexpectedly switches between different phases. Geometrically motivated metrics are developed to measure phase transitions. Finally, we show that phase transitions can be sped up with more robust augmentations. Code and visualizations will be made public upon publication.

## 1 INTRODUCTION

One of the key problems in modern machine learning is crafting effective representations of data without human-generated labels. Contrastive learning and other self-supervised learning methods are among the most popular and effective methods to date for tackling this problem (Chen et al., 2020; Chen & He, 2021; Grill et al., 2020; Bardes et al., 2022; Caron et al., 2021). In contrastive learning, positive examples, consisting of two augmented version of a single image, have their representations pushed together in the output space; negative examples, which are augmented versions of different images, have their representations pushed apart.

Why these models even work at all, given their open-ended objectives, is somewhat mysterious, and there have been multiple theories of self-supervised learning (HaoChen et al., 2021; Tian, 2022; Zimmermann et al., 2021; Tian et al., 2020a; 2021b). However, to understand how self-supervised models work, it is key to understand their training dynamics and the ways in which they differ from their supervised counterparts.

A basic question is whether self-supervised training dynamics are by nature continuous or discrete. Previous work on contrastive learning training dynamics has suggested a step-by-step mechanism where eigenmodes are learned one-by-one (Jing et al., 2022; Tian et al., 2021a; Simon et al., 2023). We propose a generalized understanding of this phenomenon where contrastive models undergo dis-

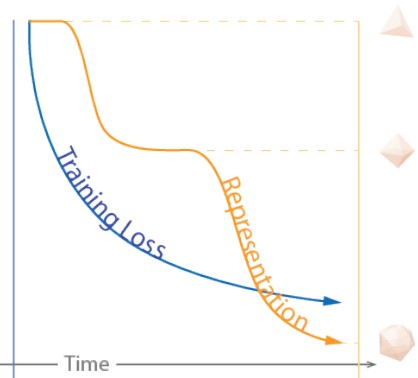

Figure 1: Loss smoothly decreases during training while the representation undergoes discrete phase transitions.

continuous *phase transitions* between discrete representation topologies or "phases". We document phase transitions which exhibit the following two key features:

1. they involve *significant* changes in the topology and geometry of the representation: what the representation actually is (as opposed to direct performance). We use visualization and tailored geometric metrics to study representations.
2. they occur *suddenly and unexpectedly* in the course of training. That a phase transition will occur can not be extrapolated from before it occurs. Between transitions, i.e. within a phase, representation topology should remain stable.

Phase transitions imply that different learned representations are emergent phenomena *in training time* (Wei et al., 2022; Nanda et al., 2023). Unlike (Simon et al., 2023), we tackle cases where training dynamics are discrete while loss continuously decreases, as we demonstrate dramatic changes in the representation geometry that are not reflected in the training loss. And by adjusting augmentation strength, we can deduce the discrete nature of phases by how representations rarely settle in intermediate forms but rather at one phase or another. Moreover, we can obeserve that the timing of transitions depends on augmentation strength. Finally, with visualization and geometric metrics, we can describe and analyze complex nonlinear transitions in fine detail (Tian, 2023).

We discuss phase transitions, related behaviors, and the aforementioned properties in three settings. First, in section 2, we examine phases in a simple linear model, where we show that the number of phases increases with the dimensionality of augmentations and provide an example of striking *non-monotonicity* in training. Second, in section 3, we examine phases in two physics-inspired toy datasets, giving a clear example of a dramatic *topological* phase transition. Finally, in section 4, we examine phase transition behavior in a ResNet-50 model trained on ImageNet with some modifications. It is well known that contrastive representations exhibit clustering behavior (Böhm et al., 2022), but the dynamics behind this capability provide evidence for our phase transition hypothesis.

Overall, we propose a new understanding of self-supervised training dynamics as involving phase transitions between discrete phases, which can be distinguished based on topological differences in representation geometry. Our perspective is that, as opposed to continuously encouraging invariances, the role of augmentations in self-supervised learning is to speed up these discrete phase transitions.

## 2 PHASES IN A LINEAR NETWORK

We first show the existence of phase behavior in a simple setting with a single-layer linear network. The earlier exposition overlaps with previous work (e.g. there are many similarities with (Simon et al., 2023)), but we culminate in an analysis of a *cosine* metric which showcases the additional complexity of self-supervised over supervised training dynamics even in simple settings. With specific initializations, phase transitions are revealed. More complex topological changes do not occur because of the simplicity of the linear setting, but there remain a number of commonalities, detailed at the end of this section.

### 2.1 PRELIMINARIES

Consider a scenario in which we are trying to contrastively learn target vectors $\mathbf{t}_1, \mathbf{t}_2, \ldots, \mathbf{t}_s \in \mathbb{R}^n$. Let our input be $\mathbf{x} \sim \mathcal{N}(0, I_n)$ where $n$ is the input and output dimension and the weights of our network be $\mathbf{W}_t \in \mathbb{R}^{n \times o}$, where $o \in \{1, n\}$ is the output dimension. We use linear augmentations $\mathbf{d}$ sampled from a distribution $p_{\mathbf{d}}(\mathbf{d})$, such that all vectors $\mathbf{d}$ within the support of $p_{\mathbf{d}}$ are orthogonal to $\mathbf{t}_i$ for all $i$.

We use variants of the InfoNCE loss with a Euclidean distance metric, temperature 2, and either a single negative example (essenetially a triplet loss) or the limit of infinite negative examples:

$$\ell = -\mathbb{E}_{\mathbf{x},\mathbf{d},\mathbf{x}'} \left[ \log \frac{e^{-\|\mathbf{W}^T(\mathbf{x}+\mathbf{d})-\mathbf{W}^T\mathbf{x}\|^2/2}}{e^{-\|\mathbf{W}^T\mathbf{x}'-\mathbf{W}^T\mathbf{x}\|^2/2}} \right] \qquad \ell_\infty = -\mathbb{E}_{\mathbf{x},\mathbf{d}} \left[ \log \frac{e^{-\|\mathbf{W}^T(\mathbf{x}+\mathbf{d})-\mathbf{W}^T\mathbf{x}\|^2/2}}{\mathbb{E}_{\mathbf{x}'}[e^{-\|\mathbf{W}^T\mathbf{x}'-\mathbf{W}^T\mathbf{x}\|^2/2}]} \right]$$

Here, the negative example $\mathbf{x}'$ is drawn from the same distribution as $\mathbf{x}$, while the positive example is $\mathbf{x} + \mathbf{d}$. We use a *gradient flow* setup, as in (Hua et al., 2021; Tian et al., 2020b; Simon et al., 2023;

Tian et al., 2021a) where the step size becomes infinitesimal and thus $\mathbf{W}_t$ changes with time as

$$d/dt\, \mathbf{W}_t = -\beta\, \nabla_{\mathbf{W}_t}\ell,$$

where $\beta$ is the instantaneous learning rate.

**Proposition 1.** There exists $\mathbf{b}_1, \mathbf{b}_2, \ldots, \mathbf{b}_m$ such that

1. $m$ is at most $n - \dim \mathrm{span}(\mathbf{t}_i)$;
2. $\mathbf{b}_i$ has $i$ nonzero coordinates;
3. $\mathbf{W}_t$ behaves identically whether we sample $\mathbf{d}$ from $p_{\mathbf{d}}$ or $p_{\mathbf{b}}$, where $p_{\mathbf{b}}$ is the uniform distribution over $\mathbf{b}_i$, for both losses $\ell$ and $\ell_\infty$.

*Proof.* See subsection A.1.

**Proposition 2.** Both $\mathbf{W}_t$ and $\ell$ (not $\ell_\infty$) experience exponential decay whose rate is, in some sense, proportional to $\mathbb{E}[\mathbf{d}\mathbf{d}^T]$.

1. $\mathbf{W}_t = \mathbf{M}_t \mathbf{W}_0$ for some $\mathbf{M}_t$, which is equal to the matrix exponential $e^{-2\beta t \mathbf{B}\mathbf{B}^T/n}$ times a non-constant scalar.
2. $\ell_t$ (the loss at time $t$) is of the form

$$\sum_{i=1}^{m} \mathrm{sgn}(\lambda_i - 2) \cdot c_i e^{2\beta t \cdot (2 - \lambda_i)}$$

for some constants $c_i \geq 0$ where $\lambda_i \geq 0$ are the eigenvalues of $\mathbb{E}[\mathbf{d}\mathbf{d}^T]$.

*Proof.* See subsection A.2.

## 2.2 THE COSINE METRIC

Let $\mathbf{B} \in \mathbb{R}^{n \times n}$ be a matrix with columns $\mathbf{b}_i$ or zeros. We can define the *cosine* between $\mathbf{W}$ and $\mathbf{B}$ as

$$\cos(\mathbf{W}, \mathbf{B}) = \frac{\langle \mathbf{W}, \mathbf{B} \rangle}{\sqrt{\langle \mathbf{W}, \mathbf{W} \rangle \langle \mathbf{B}, \mathbf{B} \rangle}},$$

where the inner product is $\langle A, B \rangle = \mathrm{tr}(A^T B)$ if $A, B$ are both matrices, $\langle a, B \rangle = \|aB\|^2$, if $a$ is a vector and $B$ a matrix (not extremely well-defined), and $\langle a, b \rangle$ is the normal dot product if $a, b$ are both vectors. This is a matrix analogue of the typical cosine $\cos(\mathbf{w}_i, \mathbf{d})$; it is equal to the cosine between vectorized $\mathbf{W}$ and $\mathbf{B}$.

**Proposition 3.** Let $\mathbf{T}$ be an $n \times n$ matrix with columns $\mathbf{t}_i$ and zeros otherwise. Define a supervised loss as

$$\ell_{\mathrm{sup}} = \mathbb{E}_{\mathbf{x}}\left[\left\|\mathbf{W}^T\mathbf{x} - \mathbf{T}^T\mathbf{x}\right\|^2\right].$$

The corresponding cosine metric $\cos_t(\mathbf{W}_t, \mathbf{T})$ is monotonic and goes to $1$ as $t \to \infty$.

*Proof.* See subsection A.3.

**Theorem.** This cosine is a function of time, of the form

$$\cos_t^2(\mathbf{W}_t, \mathbf{B}) = \left(\sum_{i=1}^{m} c_i^1 \cdot e^{-\lambda_i \cdot \beta t}\right)^2 \bigg/ \left(\sum_{i=1}^{m} c_i^2 \cdot e^{-2\lambda_i \cdot \beta t}\right),$$

where $\lambda_i \geq 0$ are the eigenvalues of $\mathbb{E}[\mathbf{d}\mathbf{d}^T]$, with multiplicity, with $\lambda_i$ decreasing, and $c_i^2 \geq 0$. Additionally, when $\lambda_i = 0$, $c_i^1 = 0$, so $\cos_\infty \to 0$.

*Proof.* See subsection A.4.

## 2.3 EIGENVALUES AND PHASE BEHAVIOR

Intuitively, for any $a, b, c, d$, where $c, d \geq 0$, we have $\min\left(\dfrac{a}{c}, \dfrac{b}{d}\right) \leq \dfrac{a+b}{c+d} \leq \max\left(\dfrac{a}{c}, \dfrac{b}{d}\right)$.

Thus, non-rigorously, we can consider $\cos_t$ to be an average of the the values $c_i^1/\sqrt{c_i^2}$ with weight $|c_i^1| e^{-\lambda_i \cdot \beta t}$.

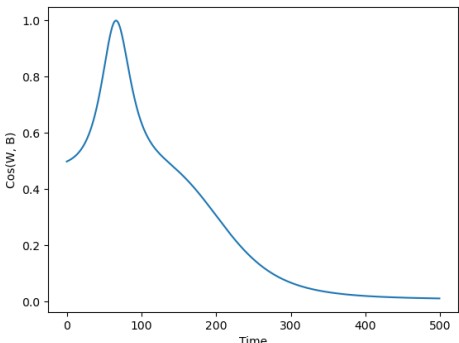

Figure 2: An example of $\cos_t$ being followed through time which is non-monotonic, $\beta = 0.01$.

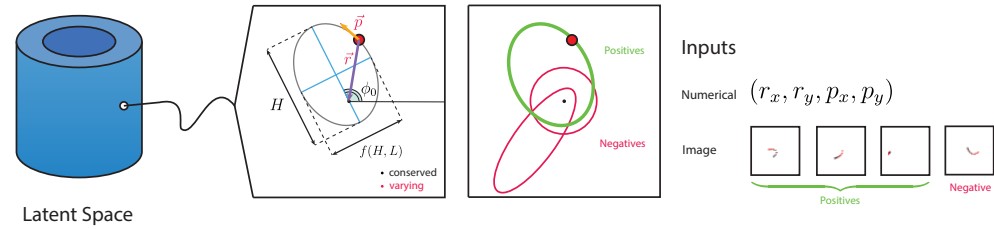

Figure 3: How the data generation for our Kepler dataset works. Images on bottom right are real examples sourced from the dataset.

There are no topological phase transitions due to the linear nature of the problem, but we can still identify phase-like behavior. One key property of phases is that they should be stable; the easiest way for $\cos_t$ to be stable is if the same exponential term is temporarily dominating in both the numerator and denominator, at which point $\cos_t \approx c_i^1 / \sqrt{c_i^2}$.

Thus, loosely, we can consider $\cos_t$ to have $m$ phases, one corresponding to each eigenvalue of $\mathbb{E}[\mathbf{d}\mathbf{d}^T]$. For all $m$ of these phases to distinctly occur and be stable for some length of time, we need $c_i \gg c_{i+1}$ for all $i$, which requires different rows of $\mathbf{W}_0$ to be of massively different magnitude, which may be unlikely in practice.

Finally, there is no restriction that $c_i^1 / \sqrt{c_i^2} > c_{i+1}^1 / \sqrt{c_{i+1}^2}$ which means phase $i$ is not necessarily a higher cosine value than $i+1$. In fact, these phases can pretty much take on any relative order, except for the final phase required to be 0 (as long as $m < n$). Thus, we can get *non-monotonic* training; see Figure 2.

## 3 PHASES IN PHYSICS-INSPIRED TOY DATASETS

### 3.1 PHYSICS DATASETS

We use two physics datasets from (Lu et al., 2023): Kepler and double pendulums.[1] In both datasets, the problem is to extract the *conserved quantities* of a dynamical system, which are properties of the system that remain constant as the system evolves. Each trajectory in our dataset is considered to be a single class and the augmentation consists of sampling from the same trajectory at different points in time. The contrastive loss then encourages temporal invariance; ideally the learned representations are completely parameterized by the conserved quantities. In the vision setting, time augmentation is also used to contrastively learn from video (Dave et al., 2022; Qian et al., 2021).

---

[1]We present our experiments on double pendulums in the Supplemental Materials.

**Kepler problem** We demonstrate the dataset generation in Figure 3. Given the position and momentum of a planet orbiting a star, three conserved quantities can be extracted. Kepler's equations describe the dynamics of this system. Three conserved quantities completely describe a trajectory; for example, we can use energy $H$, angular momentum $\|\mathbf{L}\|$, and the angle of the Laplace-Runge-Lenz vector $\phi_0$. These three conserved quantites can be calculated from the position $\mathbf{r}$ and momentum $\mathbf{p}$ with

$$H = \frac{\|\mathbf{p}\|^2}{2} - \frac{1}{\|\mathbf{r}\|} \quad \mathbf{L} = \mathbf{r} \times \mathbf{p} \quad \phi_0 = \arg(\mathbf{p} \times \mathbf{L} - \hat{\mathbf{r}})$$

where $\arg(\mathbf{v})$ is the angle of vector $\mathbf{v}$ with respect to the positive $x$-axis.

**Double pendulums.** Double pendulums yielded similar results to the Kepler dataset; we thus place the discussion in Appendix B.

## 3.2 VARYING AUGMENTATION ROBUSTNESS

Our augmentations are *non-destructive*, that is, all augmentations are between members of precisely the same class, and no meaningful information is made invariant. Because this allows us to more tightly control our experiments, this is a feature we preserve in our ImageNet experiments as well. (Alternatively, experiments indicated purely destructive augmentations result in a shrinking of the representation.) We thus refer to augmentation strength as *augmentation robustness* in order to refer to the fact that all such augmentations are non-destructive.

In order to demonstrate the effect of augmentation robustness on transition time, we weaken the temporal augmentation by sampling from a partial trajectory. Two points sampled from distant points within the trajectory provide a more robust augmentation, so sampling from a *partial* trajectory weakens augmentations.

We use a hyperparameter $\alpha$ to quantify the augmentation robustness. In this case, we sample positives from a proportion of $\alpha$ of the trajectory. If $T$ be the period of the trajectory, we randomly sample a starting time $t$ from $[0, T]$ uniformly where $T$ is the period of the orbit. Then we sample positive examples uniformly from the range $[t, t + \alpha T]$. Note that $\alpha = 1$ corresponds to full trajectories.

## 3.3 TRAINING LOSSES AND METRICS

**Contrastive loss** We use the SimCLR framework outlined by Chen et al. (2020) with variants of the InfoNCE loss (Oord et al., 2018). Given an encoder network $f(x)$ and a batch of $b$ inputs, $\{(x_i^1, x_i^2)\}_{1 \le i \le b}$, where $x_i^1$ and $x_i^2$ as two randomly augmented versions of the same data, which together form a positive sample. The loss can be written as

$$\ell = -\mathbb{E}_i \left[ \log \frac{e^{-\|f(x_i^1) - f(x_i^2)\|}}{e^{-\|f(x_i^1) - f(x_i^2)\|} + \sum_{j \ne i} e^{-\|f(x_i^1) - f(x_j^2)\|}} \right],$$

where we use Euclidean distance.

$R^2$ **metric** Our metric should compare the geometric similarity with the optimal representation, which is an affine transform of the continuous latent space (Zimmermann et al., 2021). Thus, given that we know the latent space and the parameterization used to generate our data, the quality of a linear regression from our learned representations to the latent variable(s) reflects the quality of our representation. We calculate the $R^2$ of a linear regression from the representation to the last-learned linear latent variable, as the order of learning for latents tends to be stable across initializations.

**Training details** For numerical experiments on the Kepler and double pendulum datasets, we use multilayer perceptrons with four hidden layers of width 64 and RELU activation trained with Euclidean $\delta$. We use a batch size of 512, temperature 0.5, representation size 3 and no projector. For Kepler, we have 10,240 generated trajectories with 10 time samples each, and for the double pendulum dataset we have 1,000 generated trajectories with 16 time samples each.

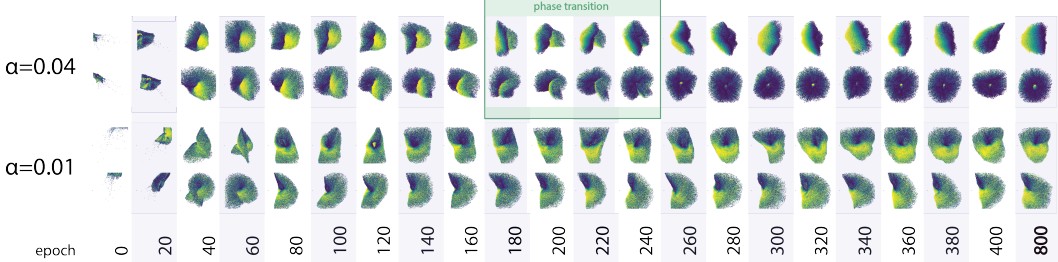

Figure 4: A phase transition with $\alpha = 0.04$ and no phase transition with $\alpha = 0.01$.

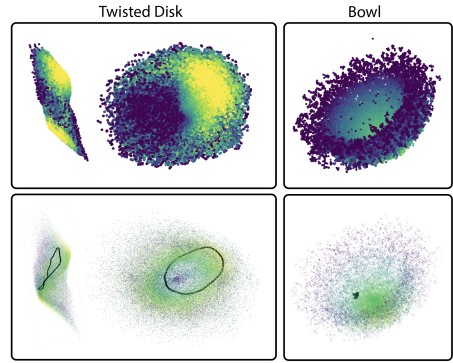

Figure 5: Visual comparison showing poor alignment in Twisted Disk, but not in the Bowl. Black outline is embedding of a single orbit.

Figure 6: Number of epochs needed until the phase transition for Kepler charted against percentage of trajectory seen during training. 16 trials for each $\alpha$ plotted, with a 95% confidence interval shown.

## 3.4 RESULTS

Because the Kepler dataset has three conserved quantities, we are able to directly visualize representations. When $\alpha = 1.0$ on the Kepler dataset, all three conserved quantities are present in the representation, and moreover, the representation correctly represents the geometry of the latent space $(\phi_0, H, L)$ in nontrivial ways. We have nicknamed this particular representation the *Bowl* because of its shape. A more detailed discussion of this representation is in Appendix C.

We can weaken the robustness $\alpha$ down to around $0.02$ and the fully trained representation, other than degrading somewhat in quality, remains roughly the same. However, when we set $\alpha \leq 0.01$, the shape of the fully trained representations looks radically different. The representations remain stuck at a shape that we call the *Twisted Disk*, which embeds position directly and has a twist at the center at which orbits with $L \approx 0, 1$ are embedded close together (Figure 5). Despite being able to achieve low training loss, the disk is misleading about the global structure of the data, and tends to have worse than random loss when re-evaluated on full trajectories. Visually, orbits maintain their elliptical shape; the disk is essentially a direct embedding of the input space.

The critical range is when $\alpha$ is near, but above, $0.01$. Instead of converging onto an intermediate form, we observe that the model *always* first learns the twisted disk, is stable for some length of time (depending on $\alpha$, see Figure 6), and then transitions into the bowl (Figure 4). The time of transition correlates strongly with a sharp increase in the $R^2$ metric (Figure 7), but without any drop in the training loss. This transition is not merely an affine transformation but involves topological reconfiguration: the central point in the disk has no analogue in the bowl. This either-or convergence highlights the discrete nature of phases.

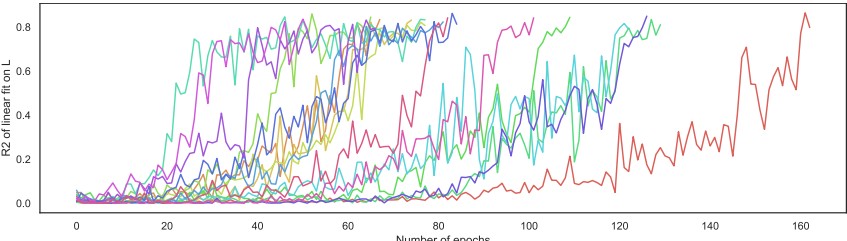

Figure 7: Tracking $R^2$ value for linear fit of network outputs on a randomly generated validation dataset of 1024 orbits on conserved quantity $L$ during the training process over training with $\alpha = 0.05$. 16 fresh runs (training and validation datasets) were tracked and logged until they satisfied $\text{avg}(R^2_{t-3:t}) > 0.8$. Note sudden, sharp increases.

## 4 PHASES IN COMPUTER VISION

Next, we study phase transitions in ImageNet (Deng et al., 2009), a standard benchmark for evaluation of contrastive learning and other computer vision methods. After training ResNet-50 with our supervised contrastive loss with various $\alpha$ (see below), we analyze the representation of a randomly-sampled 10% selection of the ImageNet training set covering all 1000 classes through visualization and a variety of clustering metrics. This setting poses an additional challenge due to the high dimensionality and thus the high number of phase transitions which likely occur.

Thus, instead of trying to isolate individual phase transitions, we quantitatively test two implications of our hypothesis. First, there should exist metrics which increase dramatically late in training in accordance with late-training phase transitions. Second, as phase transition timing depends on augmentation robustness, decreasing augmentation robustness should delay the increase of those metrics. In addition to testing these two hypotheses, we also visually inspect the learned embeddings.

### 4.1 TRAINING LOSSES AND METRICS

**Supervised contrastive learning** For our vision experiments on CIFAR-10 and ImageNet, we use supervised contrastive losses, which allows us to have a well-defined "ground truth" latent space, and also allows us to use non-destructive augmentations. In order to isolate the effect of the supervision augmentation, we do not use typical such as cropping, color jittering, flipping, etc.

We generalize the loss from Khosla et al. (2020) in order to control augmentation robustness by selecting the closest $\alpha$ members of the class. $\alpha$ is a hyperparameter which controls augmentation robustness; at $\alpha = 1.0$ we have supervised contrastive learning, and at $\alpha = 0.0$ there are no positive examples and the data points are spread evenly across the hypersphere. We bootstrap the similarity metric by using the cosine distance between the current predicted representations, which allows easy implementation.

Formally, let $S^\alpha$ denotes the $\alpha$th quantile of a multiset $S$, $P(i)$ denotes all the members of the batch with the same class as $i$ (positive examples), and $\mathcal{D}(S, a)$ be the set $\{-\delta(a, s) : s \in S\}$. Then we can write $P_\alpha(i)$, the $\alpha$ closest positives of $x_i$, as the set of $p \in P(i)$ such that $\delta(f(x_i), f(x_p)) < \mathcal{D}^\alpha(P(i), x_i)$. Our loss then is

$$\ell = -\mathbb{E}_i \left[ \frac{1}{|P(i)| \cdot \alpha} \cdot \sum_{p \in P_\alpha(i)} \log \frac{e^{-\delta(f(x_i), f(x_p))}}{\sum_{j \neq i} e^{-\delta(f(x_i), f(x_j))}} \right],$$

where $\delta$ is cosine distance. Because of the bootstrapped similarity, this loss has some unique properties discussed in Appendix D. We do not propagate gradients through the similarity metric.

For datasets with many classes like ImageNet in which each batch contains a small number of positives to begin with, the choice of how quantiles are rounded up or down matter. We round up or down probabilistically such that in expectation the number of positives is exactly $\alpha|P(i)|$.

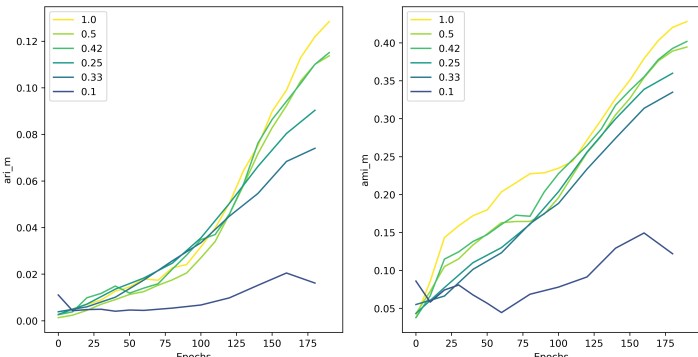

Figure 8: Tracking adjusted Rand index (ARI) and adjusted mutual information (AMI) through ImageNet training with different values of $\alpha$, plotted with moving average. Note the sudden jump in ARI around epoch 125, and how the jump is delayed for $\alpha = 0.25$.

**Clustering metrics** Unlike the physics datasets, vision datasets are categorical with representations constrained to lie on the surface of a hypersphere. The ideal representation is a high-dimensional simplex with each point corresponding to one class (Graf et al., 2021). Because our representations are constrained to be low-dimensional for visualization purposes, this is not achievable. Despite this, we can observe regular polyhedral representations in CIFAR-10 (see Appendix D).

On ImageNet, the large number of classes means we instead observe superclusters of similar classes on the sphere surface. We cluster the normalized representation with DBSCAN (Ester et al., 1996) and use the clustering metrics adjusted Rand index (ARI) and adjusted mutual information (AMI) (Vinh et al., 2010). In order that our specific choice of the sensitive DBSCAN hyperparameter $\varepsilon$ did not affect results, we conduct a hyperparameter search for $\varepsilon \in [0.005, 0.015]$ for every individual representation. We exclude any DBSCAN outliers.

**Training details** We use a projection dimension of 3 for visualization, and we visualize examples from the training set in order to see the model's learned representation, as opposed to the test embeddings which may be distorted. We use a ResNet-50 encoder (He et al., 2016), batch size 4096, learning rate 4.8 with cosine decay, weight decay 1e-6, temperature 0.2, and LARS optimizer.

## 4.2 Results

Consistent with our predictions, the ARI shows a sudden jump at around epoch 125, and decreasing $\alpha$ appears to slow down when the jump occurs (at least epoch 150 for $\alpha = 0.25, 0.33$, and not present for $\alpha = 1.0$). On the other hand, the AMI does not show any sudden jumps. It may be measuring phase transitions that occur both early and late in training.

We can also visualize the representations directly to inspect their geometry. We can see how smaller, semantically meaningful clusters separate as training proceeds; as $\alpha$ is decreased, this mechanism appears to be slowed down (Figure 9). Late in training with high $\alpha$, we can see more small clusters corresponding to more fine-grained semantic groups, which does not occur for lower $\alpha$.

## 5 Related Work

**Self-supervised training dynamics** The most closely related prior work is (Simon et al., 2023), which also investigates discrete components within self-supervised training dynamics. Unlike their work, we do not focus on linearizations of the training dynamics, but try to quantitatively and qualitatively understand nonlinear and topological changes in representations learned with deep networks. Other works that focus on linearizations include (Tian et al., 2020b; Hua et al., 2021). On the other hand, (Tian, 2022; 2023) show that nonlinear training dynamics already differ dramatically from

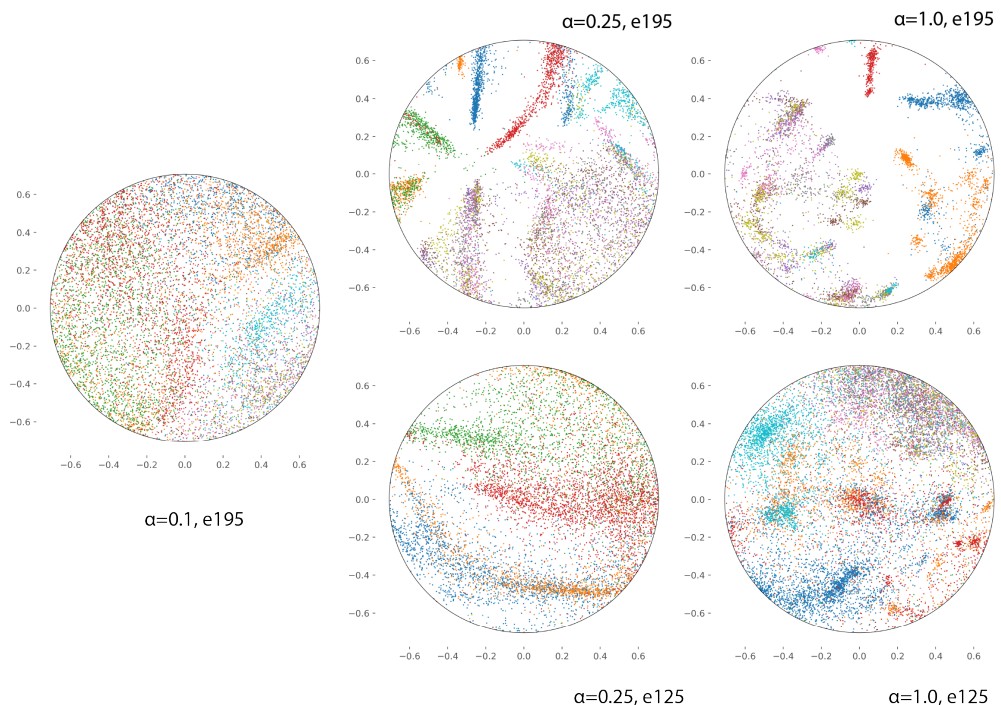

Figure 9: Direct visualization of ImageNet representations. Colored by hundreds digit of class; similar classes tend to be close numerically, creating semantically meaningful colored clusters. The projection of a $45°$ cone around a point is shown.

linear counterparts on shallow networks. As deep nonlinear networks are difficult to model theoretically, our empirical approach lets us draw new conclusions about self-supervised training dynamics.

**Augmentation strength and quality of representations** There is a "sweet spot" in terms of augmentation strength which is optimal performance on a downstream task (Tian et al., 2020a). As augmentations become too strong, representations become invariant to meaningful features. Several methods (Xiao et al., 2021; Wang & Qi, 2021; Dangovski et al., 2022) attempt to remedy this by weakening the strict invariances imposed by the contrastive loss. On the other hand, (Wang et al., 2022) argued for the usefulness of this property of contrastive learning. However, we focus on the role of augmentation robustness exclusively in the context of training dynamics.

**Deep learning to uncover conserved quantities** The physics datasets we work with involve extracting a conserved quantity from a dynamical system by learning from trajectory data. Prior work on this problem used a variety of approaches, including manifold learning combined with symbolic regression (Liu & Tegmark, 2021), manifold identification in a known symplectic geometry (Mototake, 2021), manifold learning with an optimal transport metric (Lu et al., 2023), and regression with Siamese neural networks (Wetzel et al., 2020). Identifying conserved quantities in known physical systems provides a well-controlled but complex environment for our experiments.

## 6 CONCLUSION

We analyze nonlinear, topological phase transitions between discrete phases in a linear network, physics-inspired toy datasets, and ImageNet. An extension of our results to more self-supervised learning methods, especially non-contrastive ones like SimSiam, BYOL, and others (Chen & He, 2021; Grill et al., 2020), would be valuable. For example, our insights may also apply to non-contrastive methods, whose ability to avoid collapse has been extensively analyzed (Wen & Li, 2022; Zhuo et al., 2023; Tian et al., 2021a). Stronger theoretical justifications for the occurrence of phase transitions would also be an exciting avenue for future work.

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

# A PROOFS

## A.1 PROOF OF PROPOSITION 1

For $l$, observe that

$$
\begin{aligned}
\ell &= -\mathbb{E}_{\mathbf{d}}\left[\log e^{-\left\|\mathbf{W}^T\mathbf{d}\right\|^2/2}\right] + \mathbb{E}_{\mathbf{x},\mathbf{x}'}\left[\log e^{-\left\|\mathbf{W}^T\mathbf{x}-\mathbf{W}^T\mathbf{x}'\right\|^2/2}\right] \\
&= \mathbb{E}_{\mathbf{d}}(\left\|\mathbf{W}^T\mathbf{d}\right\|^2/2) - \mathbb{E}_{\mathbf{x},\mathbf{x}'}[\left\|\mathbf{W}^T(\mathbf{x}'-\mathbf{x})\right\|^2/2] \\
&= \mathbb{E}_{\mathbf{d}}[\mathrm{tr}(\mathbf{W}^T\mathbf{d}\mathbf{d}^T\mathbf{W})/2] - \|\mathbf{W}\|^2 \\
&= \mathrm{tr}(\mathbf{W}^T\mathbb{E}[\mathbf{d}\mathbf{d}^T]\mathbf{W})/2 - \|\mathbf{W}\|^2,
\end{aligned}
$$

where we use the Frobenius norm for matrices.

For $l_\infty$, observe that

$$
\begin{aligned}
\ell_\infty &= -\mathbb{E}_{\mathbf{d}}\left[\log e^{-\left\|\mathbf{W}^T\mathbf{d}\right\|^2/2}\right] + \mathbb{E}_{\mathbf{x}}\left[\log \mathbb{E}_{\mathbf{x}'}\left[e^{-\left\|\mathbf{W}^T\mathbf{x}-\mathbf{W}^T\mathbf{x}'\right\|^2/2}\right]\right] \\
&= \mathrm{tr}(\mathbf{W}^T\mathbb{E}[\mathbf{d}\mathbf{d}^T]\mathbf{W})/2 + \mathbb{E}_{\mathbf{x}}\left[\log \mathbb{E}_{\mathbf{x}'}\left[e^{-\left\|\mathbf{W}^T\mathbf{x}-\mathbf{W}^T\mathbf{x}'\right\|^2/2}\right]\right].
\end{aligned}
$$

Observe that the dependence on $\mathbf{d}$ is only through $\mathbb{E}[\mathbf{d}\mathbf{d}^T]$. For any $\mathbf{d}_i \in \mathrm{supp}(p_{\mathbf{d}})$, $\mathbf{d}_i\mathbf{d}_i^T$ is a symmetric positive semidefinite matrix. Thus $\mathbb{E}[\mathbf{d}\mathbf{d}^T]$ is also symmetric positive semidefinite, so it has a Cholesky factorization $n\mathbb{E}[\mathbf{d}\mathbf{d}^T] = \mathbf{B}\mathbf{B}^T$, where the rank of $\mathbf{B}$ is equal to the rank of $\mathbb{E}[\mathbf{d}\mathbf{d}^T]$.

We claim the nonzero columns $\mathbf{b}_i$ satisfy all three properties. First, since $\mathbf{d}_i$ are perpendicular to $\mathbf{t}_j$, this rank is at most that of $\mathbb{R}^n \perp \mathrm{span}(\mathbf{t}_i)$, thus $m \le n - \dim \mathrm{span}(\mathbf{t}_i)$. The second fact is given by the Cholesky factorization. The third fact comes from

$$
\mathbb{E}[\mathbf{b}\mathbf{b}^T] = \frac{1}{n}\sum_{i=1}^m \mathbf{b}_i\mathbf{b}_i^T = \frac{1}{n}\mathbf{B}\mathbf{B}^T = \mathbb{E}[\mathbf{d}\mathbf{d}^T].
$$

□

Assume from here onwards that if we are using the loss $l_\infty$ then $o = 1$.

## A.2 PROOF OF PROPOSITION 2

First, we consider the $\ell$ case. From the proof of Proposition 1, we have

$$
\begin{aligned}
d/dt\, \mathbf{W}_t &= -\beta \nabla_{\mathbf{W}_t}\ell \\
&= -\beta\left(\mathbb{E}[\mathbf{d}\mathbf{d}]^T\mathbf{W} - 2\mathbf{W}\right) \\
&= \beta\left(2\,\mathbf{I} - \mathbf{B}\mathbf{B}^T/n\right)\mathbf{W}.
\end{aligned}
$$

This can be solved with matrix exponentiation, i.e. the solution is

$$
\mathbf{W}_t = e^{\beta t\cdot(2\mathbf{I}-\mathbf{B}\mathbf{B}^T/n)}\mathbf{W}_0.
$$

Then the loss is

$$
\begin{aligned}
\ell_t &= \mathrm{tr}(\mathbf{W}_0^T e^{2\beta t\cdot(2\mathbf{I}-\mathbf{B}\mathbf{B}^T/n)}\mathbb{E}[\mathbf{d}\mathbf{d}^T]\mathbf{W}_0) - \mathrm{tr}(\mathbf{W}_0^T e^{2\beta t\cdot(2\mathbf{I}-\mathbf{B}\mathbf{B}^T/n)}\mathbf{W}_0) \\
&= \mathrm{tr}(\mathbf{W}_0^T e^{2\beta t\cdot(2\mathbf{I}-\mathbf{B}\mathbf{B}^T/n)}(\mathbb{E}[\mathbf{d}\mathbf{d}^T]/2 - \mathbf{I})\mathbf{W}_0).
\end{aligned}
$$

as $A$ and $e^A$ generally commute. Observe that $\mathbb{E}[\mathbf{d}\mathbf{d}^T]$ is diagonalizable; thus see the proof of the Theorem for a very similar argument.

Now we consider $\ell_\infty$. Note that $\mathbf{W}$ is a vector here, so we denote it as $\mathbf{w}$. The key additional difficulty is computing the uniformity term

$$
\mathbb{E}_{\mathbf{x}}\left[\log \mathbb{E}_{\mathbf{x}'}\left[e^{-(\mathbf{w}^T\mathbf{x}-\mathbf{w}^T\mathbf{x}')^2/2}\right]\right].
$$

If $y \sim \mathcal{N}(\mu, \sigma)$, observe

$$\mathbb{E}[e^{y^2/2}] = \int_{-\infty}^{\infty} e^{t^2/2} \cdot e^{-(t-\mu)^2/2\sigma^2} \cdot \frac{1}{\sqrt{2\pi\sigma^2}} \, dt$$

$$= \int_{-\infty}^{\infty} \exp\left(-\left(t - \mu \cdot \frac{(1/\sigma^2 - 1)^{-1}}{\sigma^2}\right)^2 \cdot \frac{1}{2(1/\sigma^2 - 1)^{-1}} + \mu^2 \cdot \frac{1}{2(1/\sigma^2 - 1)\sigma^4} - \frac{\mu^2}{2\sigma^2}\right) \cdot \frac{1}{\sqrt{2\pi\sigma^2}} \, dt$$

$$= \sqrt{2\pi(1/\sigma^2 - 1)^{-1}} \cdot \frac{1}{\sqrt{2\pi\sigma^2}} \cdot \exp\left(\frac{\mu^2}{2(1 - \sigma^2)}\right)$$

$$= \frac{1}{\sqrt{1 - \sigma^2}} \cdot \exp\left(\frac{\mu^2}{2(1 - \sigma^2)}\right).$$

The logarithm is thus $-\frac{1}{2}\log(1 - \sigma^2) + \frac{\mu^2}{2(1-\sigma^2)}$.

Note that $\mathbf{w}^T(\mathbf{x} - \mathbf{x}')$ is normally distributed with mean $\mathbf{w}^T\mathbf{x}$ and variance $\|\mathbf{w}\|^2$. So we can apply this formula, further noting that the expected value of $(\mathbf{w}^T\mathbf{x})^2$ is also $\|\mathbf{w}\|^2$. Overall, this implies we can write this term as a function $g$ of $\|\mathbf{w}\|^2$.

Taking the derivative, we thus get

$$d/dt \, \mathbf{w}_t = -\beta \, \nabla_{\mathbf{w_t}} \ell$$

$$= -\beta \left(\mathbb{E}[\mathbf{dd}]^T \mathbf{w} - 2g'(\|\mathbf{w}\|^2)\mathbf{w}\right)$$

$$= \beta \left(2g'(\|\mathbf{w}\|^2)\mathbf{I} - \mathbf{BB}^T/n\right) \mathbf{w}.$$

Observe that although the matrix factor $2g'(\|\mathbf{w}\|^2)\mathbf{I} - \mathbf{BB}^T/n$ changes with time, all such matrix factors from different times commute. Thus, using the Magnus expansion, we get

$$\mathbf{w}_t = e^{\beta \cdot \left(\left(\int_0^t 2g'(\|\mathbf{w}_s\|^2)ds\right)\mathbf{I} - t\mathbf{BB}^T/n\right)}\mathbf{w}_0 = e^{\beta \cdot \left((g(\|\mathbf{w}_t\|^2) - g(\|\mathbf{w}_0\|^2))\mathbf{I} - t\mathbf{BB}^T/n\right)}\mathbf{w}_0.$$

$\square$

### A.3 Proof of Proposition 3

The loss is equal to

$$\text{tr}((\mathbf{W} - \mathbf{T})^T(\mathbf{W} - \mathbf{T})) = \|\mathbf{W}\|^2 + \|\mathbf{T}\|^2 - 2\langle\mathbf{W}, \mathbf{T}\rangle.$$

The gradient of this with respect to $\mathbf{W}$ is $2(\mathbf{W} - \mathbf{T})$. Thus $\mathbf{W} - \mathbf{T}$ experiences exponential decay with a rate of $-2\beta$, i.e.

$$\mathbf{W}_t = \mathbf{T} + e^{-2\beta t}(\mathbf{W}_0 - \mathbf{T}).$$

Next, we calculate the numerator squared:

$$\langle\mathbf{W}_t, \mathbf{T}\rangle^2 = (\|\mathbf{T}\|^2 + e^{-2\beta t}(\langle\mathbf{W}_0, \mathbf{T}\rangle - \|\mathbf{T}\|^2))^2$$

$$= \|\mathbf{T}\|^4 + 2e^{-2\beta t}\|\mathbf{T}\|^2(\langle\mathbf{W}_0, \mathbf{T}\rangle - \|\mathbf{T}\|^2) + e^{-4\beta t}(\langle\mathbf{W}_0, \mathbf{T}\rangle - \|\mathbf{T}\|^2)^2.$$

Then $\langle\mathbf{W}, \mathbf{W}\rangle$:

$$\langle\mathbf{W}_t, \mathbf{W}_t\rangle = \|\mathbf{T}\|^2 + 2e^{-2\beta t}(\langle\mathbf{W}_0, \mathbf{T}\rangle - \|\mathbf{T}\|^2) + e^{-4\beta t}\|\mathbf{W}_0 - \mathbf{T}\|^2.$$

We discuss this point more in the main text, but essentially, after accounting for the $\|\mathbf{T}\|^2$ term in the denominator, this is some average (not necessarily arithmetic average) of the ratios

$$\frac{\|\mathbf{T}\|^4}{\|\mathbf{T}^2\|} \cdot \frac{1}{\|\mathbf{T}\|^2} = 1, \qquad \frac{2e^{-2\beta t}\|\mathbf{T}\|^2(\langle\mathbf{W}_0, \mathbf{T}\rangle - \|\mathbf{T}\|^2)}{2e^{-2\beta t}(\langle\mathbf{W}_0, \mathbf{T}\rangle - \|\mathbf{T}\|^2)} \cdot \frac{1}{\|\mathbf{T}\|^2} = 1,$$

$$\frac{\langle\mathbf{W}_0 - \mathbf{T}, \mathbf{T}\rangle^2}{\|\mathbf{W}_0 - \mathbf{T}\|^2} \leq 1$$

i.e. the ratios between the coefficients of the same exponentials in the numerator and denominator. The last term, with "weight" $e^{-4\beta t}$, is comparatively favored for small $t$ but decays to zero weight as $t \to \infty$. This thus gives us monotonically increasing, as desired. $\square$

### A.4 PROOF OF THEOREM

Observe that $\cos(\mathbf{W}, \mathbf{B})$ doesn't change if $\mathbf{W}$ is multiplied by any scalar factor. So we can use a simpler $\mathbf{W}_t = e^{-t\beta/n\mathbf{BB}^T}\mathbf{W}_0$. Next, observe that $\langle \mathbf{B}, \mathbf{B} \rangle$ is a constant factor and thus can be absorbed into the $c_i^1$s, meaning that we only need to compute

$$\frac{\langle \mathbf{W}, \mathbf{B} \rangle^2}{\langle \mathbf{W}, \mathbf{W} \rangle}, \qquad \mathbf{W} = e^{-t\beta/n\mathbf{BB}^T}\mathbf{W}_0.$$

Expanded out with the definition of the inner product and the solution for $\mathbf{W}$, we get

$$\frac{\text{tr}(\mathbf{W}_0^T e^{-t\beta/n\mathbf{BB}^T}\mathbf{B})^2}{\text{tr}(\mathbf{W}_0^T e^{-2t\beta/n\mathbf{BB}^T}\mathbf{W}_0)}.$$

Since $\mathbf{BB}^T/n = \mathbf{E}[\mathbf{dd}^T]$ is a real symmetric matrix, it can be diagonalized into $\mathbf{Q}^T \mathbf{\Lambda} \mathbf{Q}$ where $\mathbf{Q}$ is orthogonal and $\mathbf{\Lambda}$ is diagonal. Thus we can write this expression as

$$\frac{\text{tr}(\mathbf{W}_0^T \mathbf{Q}^T e^{-t\beta\mathbf{\Lambda}}\mathbf{Q}\mathbf{B})^2}{\text{tr}(\mathbf{W}_0^T \mathbf{Q}^T e^{-2t\beta\mathbf{\Lambda}}\mathbf{Q}\mathbf{W}_0)}.$$

Consider something of the form $\text{tr}(\mathbf{A}^T\mathbf{X}\mathbf{B})$ where $\mathbf{X}$ is a diagonal matrix with entries $x_i$. First,

$$(\mathbf{X}\mathbf{B})_{ij} = \sum_{k=1}^{n} \mathbf{X}_{ik}\mathbf{B}_{kj} = x_i\mathbf{B}_{ij}.$$

Then

$$(\mathbf{A}^T(\mathbf{X}\mathbf{B}))_{ii} = \sum_{k=1}^{n} \mathbf{A}_{ik}^T(\mathbf{X}\mathbf{B})_{ki} = \sum_{k=1}^{n} \mathbf{A}_{ki}x_k\mathbf{B}_{ki}.$$

Thus

$$\text{tr}(\mathbf{A}^T\mathbf{X}\mathbf{B}) = \sum_{i=1}^{n}(\mathbf{A}^T(\mathbf{X}\mathbf{B}))_{ii} = \sum_{i=1}^{n}\sum_{k=1}^{n} \mathbf{A}_{ik}^T(\mathbf{X}\mathbf{B})_{ki}$$
$$= \sum_{i=1}^{n}\sum_{k=1}^{n} \mathbf{A}_{ki}x_k\mathbf{B}_{ki} = \sum_{k=1}^{n} x_k \sum_{i=1}^{n} \mathbf{A}_{ki}\mathbf{B}_{ki}.$$

From this, we get that our expression is

$$\frac{\left(\sum_{k=1}^{n} e^{-t\beta\lambda_k} \sum_{i=1}^{n}(\mathbf{Q}\mathbf{W}_0)_{ki}(\mathbf{Q}\mathbf{B})_{ki}\right)^2}{\sum_{k=1}^{n} e^{-2t\beta\lambda_k} \sum_{i=1}^{n}(\mathbf{Q}\mathbf{W}_0)_{ki}^2}.$$

Setting the right-side summations to be $c_i^1$ and $c_i^2$, respectively, we get the desired form. When $\lambda_i = 0$, the corresponding row in $\mathbf{B}$ is zero, so $c_i^1 = 0$ as well. □

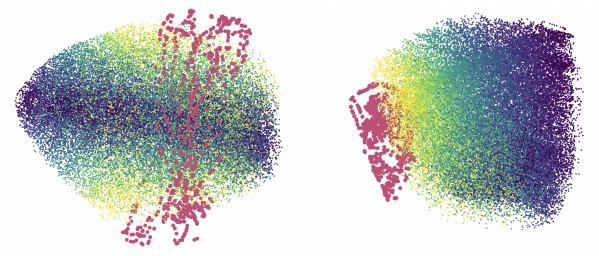

Figure 10: Embedding of low energy pendulums ($E < -1.5$) colored by $E_-$. Embeddings from a single pendulum are shown in pink. Left: radial embedding, pre-phase transition; right: linear embedding, post-phase transition.

## B    DOUBLE PENDULUM RESULTS

As a second validation of the existence of phase transitions, we study a double pendulum dataset.

Double pendulums exhibit two primary modes of behavior: at high energies, they are chaotic, whereas at lower energies they behave approximately like two coupled pendulums. At higher total energies there is only one conserved quantity $E$, while at lower total energies, the energies of each of the two pendulums $E_+ + E_- = E$ are also approximately conserved. We use the dataset open-sourced by (Lu et al., 2022).

For the double pendulum dataset, when $\alpha = 1.0$ all three conserved quantities $E, E_+, E_-$ are likewise linearly embedded and easy to read off. In the double pendulum system, when we set $\alpha < 0.1$, the representation for low energy pendulums changes considerably. While the high energy pendulums do not change, the low energy pendulums exhibit two phases. The first is the linear embedding which is most faithful to the geometry of the latent space. The second is where $E_-$ is *radially* embedded, i.e. where $E_-$ is encoded by distance from a central axis of symmetry (Figure 10). The circular motion exhibited by a single pendulum, reminscient of the pendulum's motion itself, suggests that this embedding is more direct, but less informative.

Similarly to the Kepler dataset, we can use $R^2$ to track the phase transition and plot the dependence of the phase transition's timing on robustness $\alpha$. The timing of the radial-linear phase transition happens in a fashion similar to that for the Kepler numerical dataset. We set the $R^2$ threshold at 0.6, and capped training at 1000 epochs.

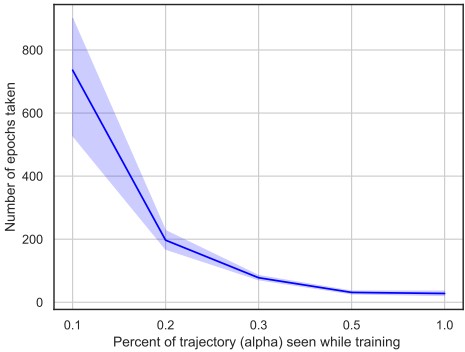

Figure 11: Phase transition diagram (analogous to Figure 6) for the double pendulum system. 16 trials for each $\alpha$ plotted, with a 95% confidence interval shown.

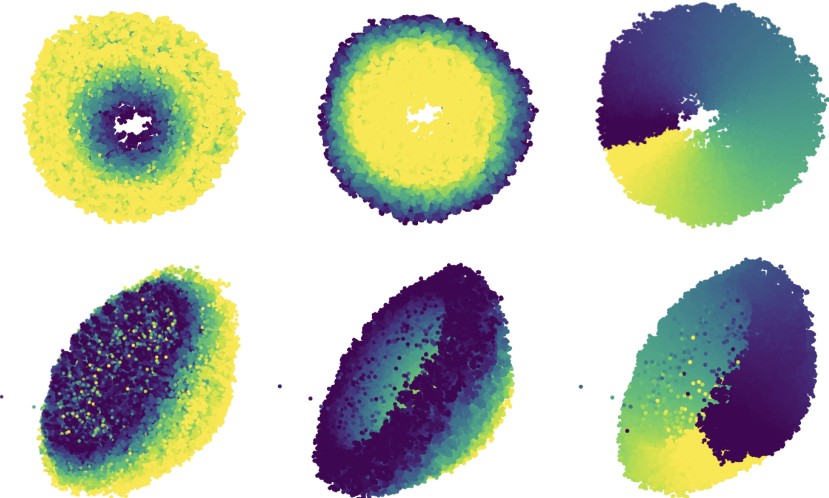

Figure 12: Bowl representation geometry obtained from a ReLU network trained over 1500 epochs presented from two different angles and colored by conserved quantities $H, L, \phi_0$ from left to right.

## C  DISCUSSION OF "BOWL" GEOMETRY

The bowl nontrivially represents the latent geometry in the following ways:

1. Orientation $\phi_0$, which is a rotational component, is encoded rotationally, so that $\phi_0 = 0, 2\pi$ are encoded to be at the same location. $H, L$ are encoded linearly.
2. The bowl has a hollow interior, which makes it a *bowl* as opposed to a *cone*. This is because of the data generation process we use; for any given momentum $L$, the range of possible energies $H$ cuts off at or above the true physical minimum, meaning that parts of the dataset are "missing".
3. The rotational axis of symmetry, about which we can define orientation $\phi_0$ (and on which $\phi_0$ is ill-defined), corresponds to the lowest given $H$ for any given $L$, which is also when the orbit is perfectly circular and thus its orientation $\phi_0$ becomes ill-defined.

We can validate this visually by plotting slices of the input space, or empirically by polynomial regression of degree 2.

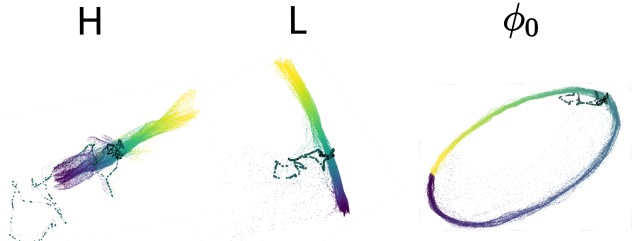

Figure 13: Portion of overall representation geometry corresponding to fixing two conserved quantities and varying the other (i.e. $H \in [-0.5, -0.25], L = 0.5, \phi_0 = \pi$ in the leftmost embedding) colored by the remaining conserved quantity. An additional single orbit (all quantities fixed - i.e. $H = -0.375, L = 0.5, \phi_0 = \pi$ in the leftmost embedding) is also plotted as denser, shaded circles.

### C.1  FIGURE-EIGHT BOWL AND OTHER RARE VARIANTS

Occasionally, when training with short trajectories, we observe a *figure-eight* variant of the bowl, which is identical to the bowl but $\phi_0$ is encoded rotationally around a figure-eight instead of a circle

(Figure 14). It appears to be an artifact of the training data; it happens particularly often (20-50%) for some datasets generated with the same general parameters.

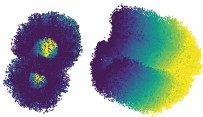

Figure 14: Two views of figure eight bowl, colored by $L$.

More rarely, we have observed other bowl variants, e.g. ones with an extremely thin $H$ axis, or flattened into 2D (dimensional collapse). These, similarly, appear to be hinged on unusual generations of the input data. They converge for very long training times to various bowl variations ($> 1000$ epochs) but will "retrain" into the normal bowl if trained on other random generations of the same input data. Additionally, shapes tend to change slightly when changing training parameters, e.g. batchnorm, although the difference tends to be minimal in most cases and topology remains the same.

## D    CIFAR-10 RESULTS

For experiments on CIFAR-10, we use a ResNet-18 encoder with a one-hidden layer projector with hidden size $2048$ and output size 3 (for visualization purposes). We use a CIFAR-specialized version of the ResNet where the first layer has kernel size 3, stride 1, and padding 1. The model is trained with our custom supervised contrastive loss and temperature $0.5$, representation size $2048$, learning rate $0.01$ and cosine decay, SGD optimizer, and mixed-precision training (Micikevicius et al., 2018).

### D.1    PHASE TRANSITIONS IN SELF-SUPERVISED CIFAR-10

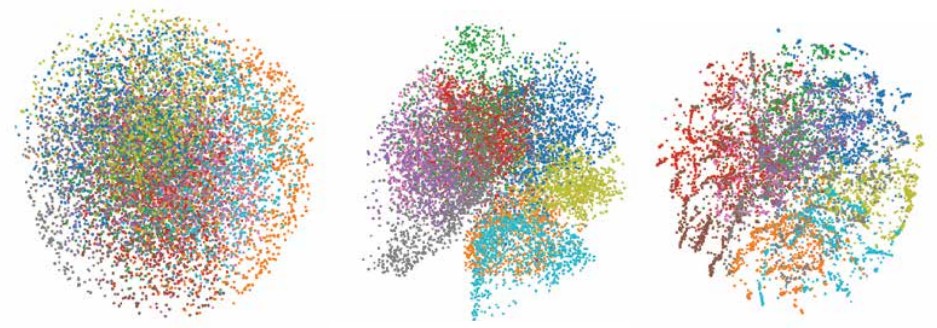

Figure 15: Geometrical training progression on projector output of models trained on CIFAR-10 with L2NCE loss; colored by class.

It is unlikely we can observe similarly clear-cut phase transitions when training on CIFAR-10 because of its complexity. However, we can observe repeatable geometric features of the representation which occur during training, which is similar in nature. We trained a self-supervised version of Sim-CLR (i.e. not supervised CL) where we forced the projector output to have output dimensionality three and used Euclidean distance with no other changes. We consistently observed three major stages as training progressed: (1) well-mixed clusters between classes; (2) class separation with noticeable margins; (3) fine-grained cluster emergence within classes (Figure 15). We believe that this appearance is consistent with a series of more local phase changes that together produce these three major phases in training.

### D.2    CLUSTERS IN SUPERVISED CIFAR-10

For a small number of clusters with a large number of samples per cluster, we can observe that the final representations produced by supervised contrastive learning are quite regular. Each class becomes its own isolated cluster and the position of the clusters are situated at the vertex of a polyhedron, e.g. for four, five, six and ten classes (Figure 16).

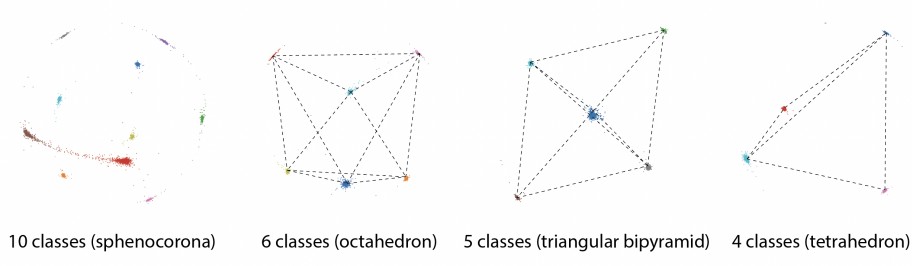

10 classes (sphenocorona)    6 classes (octahedron)    5 classes (triangular bipyramid)    4 classes (tetrahedron)

Figure 16: Polyhedral representations for a small number of classes on CIFAR-10; embeddings of the training set, normalized representations, colored by class. Note that the polyhedron for four classes is the 3-simplex as predicted (Graf et al., 2021).

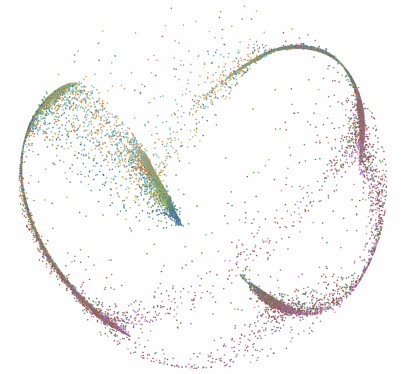

Figure 17: "Ribbon" embedding geometry for low $\alpha$ on CIFAR-10. Colored by class.

During training, we can observe large clusters first forming, and smaller clusters gradually budding off from them. This topological change constitutes a phase transition. However, with low levels of augmentation robustness, there were often clusters from different classes close together, i.e. representation quality decreased.

Because of the small number of classes, the pre-phase transition embedding is highly regular (Figure 17). The high level of symmetry in this representation is likely due to the following phenomenon. Because only $\alpha$ of each class is considered to be positive examples, the optimal embedding for the loss is actually to separate each class into approximately $1/\alpha$ clusters, as then the number of positive examples matches the number of other points within the cluster. Thus, when $\alpha$ is low, subclassification within a class is encouraged. For $\alpha \approx 0.5$ we can observe that each class in CIFAR-10 tends to be separated into two even subclusters, and there is no obvious trait differentiating the images from the two clusters. This may also explain the appearance of some of the ImageNet representations.

