# OpenReview forum: "Phase Transitions in Contrastive Learning"
_ICLR.cc/2024/Conference — Submitted to ICLR 2024_

### Official Review · Reviewer_53nW · 2023-10-24

**Soundness:** 3 good
**Presentation:** 2 fair
**Contribution:** 2 fair
**Rating:** 5
**Confidence:** 3

**Summary:**

This paper tries to justify the training dynamics of contrastive learning and measures them using geometrically motivated metrics.
Based on SimCLR, they show that the training can be split into multiple phases, with each phase having locally stable representations and different phases having rapid switches. The experiments on the linear settings, low-dimensional physics-inspired datasets and ImageNet demonstrate the existence of phase switches.

**Strengths:**

1. The new perspective using a cosine metric to describe the dynamics of the training of contrastive learning.

**Weaknesses:**

While the proposed insight for contrastive learning is interesting, I also have some concerns regarding the theory and the experiments.

1. The motivation for proposing the Theorem.  What is the extension or application if we know the phase switching in contrastive learning? Could you provide more explanation or experiments?

2. Why use the proposed metric to measure the training dynamics?  Why does the Cosine similarity increase at the time 0 - 90 in Figure 2 but then decrease? Could you provide some explanation for the reasons of increasing and decreasing in all the experiments?

3. The experiments in Figure 8 seem like not having rapid phase switching, which might imply poor soundness on deeper and larger models.

**Questions:**

Except the question of weakness, more questions can be:

1. Besides the augmentation strength, what else factors can influence the phase transitions? For example, in InfoNCE loss, we often set a hyper-parameter before exp operation to adjust the punishment strength for positive pairs.

---

### Official Review · Reviewer_KFjy · 2023-10-31

**Soundness:** 3 good
**Presentation:** 3 good
**Contribution:** 2 fair
**Rating:** 5
**Confidence:** 3

**Summary:**

This paper examines the training dynamics of contrastive learning, revealing the presence of distinct phases. Throughout the training process, the model undergoes alternating periods of locally stable representations and phase transitions, characterized by rapid changes in the learned representation geometry.

The study demonstrates the existence of these contrastive learning phases in three diverse scenarios: a simple linear model, a physics-inspired learning problem, and the ImageNet supervised contrastive learning tasks. Furthermore, the authors provide evidence that employing more potent 'non-destructive' augmentations during pretraining may accelerate the occurrence of phase transitions.

**Strengths:**

The paper offers compelling proof of the presence of distinct phases in the context of contrastive learning. I found it particularly compelling that the authors not only establish the existence of these phases but also illustrate their occurrence in three diverse settings, each varying in terms of complexity. This approach enables both a theoretical and empirical exploration of this phenomenon.

**Weaknesses:**

Although the paper effectively demonstrates the presence of phases in contrastive learning, it is not clear why those phases happen and why are those phases important? What the impact of the phase transition on the learned representation and their downstream performances?

Furthermore, it's unclear whether these phases are unique to contrastive learning or if the observations can be extended to other self-supervised and supervised approaches.

**Questions:**

I would encourage the author to better motivate the importance of phase-transition in contrastive learning. Why is this phenomenon important, and can it potentially be harnessed to overcome some of the current limitations in representation learning?

The authors argues that robust data-augmentation allow to speed-up the apparition of phase transitions. But we also know that data-augment prevents the learning of shortcut solution (such as only focusing on color distribution in images) in contrastive learning. Is there a way to relate phase-transition in the representation to the learning of specific features and how it avoids potential shortcut?

---

### Official Review · Reviewer_7pQx · 2023-11-01

**Soundness:** 2 fair
**Presentation:** 2 fair
**Contribution:** 2 fair
**Rating:** 3
**Confidence:** 4

**Summary:**

This work studies contrastive learning in three settings: 1) gaussian data with linear model, 2) physics datasets non-linear model, 3) image datasets with non-linear model. The work develops theory in the linear setting, suggesting that while the training loss may converge monotonically, the geometry of the representation space undergoes discrete phase transitions. Numerical experiments provided in all considered settings.

## Detailed Summary

### Linear setting:
* learn invariance to additive augmentation vectors, and end up with representations that span the space orthogonal to the additive augmentation vectors d… these vectors are called targets t
* assume infinitesimal step-size
* show that sampling augmentations from the space spanned by d according to some continuous distribution with support orthogonal to the target vectors is equivalent in this setting to sampling augmentations from a discrete uniform distribution over a set of basis vectors spanning the space orthogonal to the predefined targets t
* Prop3: the weight matrix converges to the matrix whose columns form a basis for the space spanned by the targets
* define a cosine metric between matrices, that is used to measure how close two matrices are to one another
* argue that while the weight matrix converges to the target matrix monotonically, it’s repulsion from the space spanned by the augmentation basis vectors is non-monotonic with the cosine measure
* No phase transition observed in numerical experiments

### Physics dataset
* consider the Kepler dataset, where each sample consists of a video of a point moving along a 3d trajectory, which can be described with three “latent” variables
* train a contrastive model where positive samples come from frames in the same trajectory and negatives come from different trajectories, parametrized by a different set of latent variables
* show than when sampling positives at short temporal horizons (i.e., “weak” augmentations), the model first learns a type of shortcut solution that exhibits a certain geometry in latent space, and then phase transitions suddenly into another geometry that captures the underlying factors of variation

### Imagenet/CIFAR10
* consider supervised contrastive learning, where "strength" of augmentations are computed by sampling images from the same class with varying visual similarity
* measure the degree of representation clustering during training with adjusted Rank Index (ARI) and adjusted Mutual Information (AMI) and find that the rate at which these metrics improve depend on the "strength" of the augmentations

**Strengths:**

### Significance
* This work studies an interesting problem targeting an improved understanding of the training dynamics of contrastive learning

### Quality
* First seek to motivate claims with a theoretical explanation in the linear settings
* Consider both small-scale (linear models with gaussian data / physics data) and larger-scale (ImageNet) problem settings

### Clarity
* Figures in the setting of the physics dataset are interesting and engaging

**Weaknesses:**

### Theory
* The theoretical intuitions derived from the theory are quite hand-wavy, even for describing how representations evolve in the toy setting. For instance, to argue how a phase could be stable, it is stated that the exponential terms must dominate the expression for the matrix cosine, which requires $cos_t \approx c^1_i / \sqrt{c^2_i}$. But, in fact, these quantities depend on a complex ratio of matrix power series, which themselves depend on the eigendecomposition of the matrix B, which itself is computed from a Cholesky factorization of the expected outer product of the additive data augmentation vectors, and this of course depends on the distribution from which augmentation vectors are sampled. When would such a criteria ever be satisfied in practice and how would you know by looking at your data or your data augmentations?
* Why do you not observe phase transformations in the linear setting? How would you re-design your problem based on your theory to observe phase transitions?

### Physics Dataset
* It is not clear to me whether observations in the physics dataset originate from the rough theoretical motivation developed in the linear setting.
* It is also not clear to me how hyper-parameter considerations (learning rate/schedules/etc.) affect any observed behaviours in terms of the geometry of the representation space.

### Image Datasets
* Quite frankly, I do not see any significant phase transition based on the ARI metric or the AMI metric.
* The low-dimensional clusterings of the image representations for supervised contrastive learning show that supervised contrastive learning clusters images form the same class together during training, however, this has already been observed in the literature, and I'm not sure that there are any new interesting intuitions about how the geometry of the representation space evolves.
* Not clear to me how the theory in the linear setting has contributed to explaining the progressive image clustering in this supervised contrastive learning setting.

**Questions:**

* While you observe a phase transition in the Physics Dataset, how can you verify that this is due to the intuition developed in the theory section?

* How robust are these observations to changes in hyper-parameters, such as learning rate magnitude or schedules. What if you reverted to the infinitesimal learning rate assumption made in the linear setting?

* How does the AMI/ARI metric relate to the convergence to the “target” matrix versus the potentially non-monotonic convergence to the B matrix in the linear setting?

---

### Meta-Review · Area_Chair_NyyC · 2023-12-11

**Metareview:**

The paper studies phase transitions in contrastive learning in three scenarios: linear setting, physics inspired dataset and on the contrastive learning of representations on the imagenet dataset.

Reviewers agreed that while it is interesting to show the existence of these phase transitions , the paper does not adress what is the impact of this on the learned representations and downstream tasks. This is an important point to motivate the study and to link  the phases in the learning to some characterization of the representation and its power in the downstream task. Reviewer KFjy
 suggested to understand the effect of data augmentation , on the phases but also on the resulting representation, whether it makes it avoid shortcuts to simple features.

Authors did not provide a rebuttal. We encourage the authors to take reviewers feedback into account and better  motivate the work in linking the phase transition in contrastive learning to characteristics of the learned representation and its impact on downstream tasks.

**Justification For Why Not Higher Score:**

The paper lacks a motivation in terms of what do phase transition in CL mean in terms of impact of the representation on downstream tasks.

**Justification For Why Not Lower Score:**

N/A

---

### Decision · Program_Chairs · 2024-01-16

Reject